# Calibrated Adaptive Teacher for Domain-Adaptive Intelligent Fault Diagnosis

**DOI:** 10.3390/s24237539

**Published:** 2024-11-26

**Authors:** Florent Forest, Olga Fink

**Affiliations:** Intelligent Maintenance and Operations Systems, École Polytechnique Fédérale de Lausanne, 1015 Lausanne, Switzerland

**Keywords:** intelligent fault diagnosis, unsupervised domain adaptation, self-training, pseudo-labels, mean teacher, calibration

## Abstract

Intelligent fault diagnosis (IFD) based on deep learning can achieve high accuracy from raw condition monitoring signals. However, models usually perform well on the training distribution only, and experience severe performance drops when applied to a different distribution. This is also observed in fault diagnosis, where assets are often operated in working conditions different from the ones in which the labeled data have been collected. The scenario where labeled data are available in a source domain and only unlabeled data are available in a target domain has been addressed recently by unsupervised domain adaptation (UDA) approaches for IFD. Recent methods have relied on self-training with confident pseudo-labels for the unlabeled target samples. However, the confidence-based selection of pseudo-labels is hindered by poorly calibrated uncertainty estimates in the target domain, primarily due to over-confident predictions, which limits the quality of pseudo-labels and leads to error accumulation. In this paper, we propose a novel method called Calibrated Adaptive Teacher (CAT), where we propose to calibrate the predictions of the teacher network on target samples throughout the self-training process, leveraging post hoc calibration techniques. We evaluate CAT on domain-adaptive IFD and perform extensive experiments on the Paderborn University (PU) benchmark for fault diagnosis of rolling bearings under varying operating conditions, using both time- and frequency-domain inputs. We compare four different calibration techniques within our framework, where temperature scaling is both the most effective and lightweight one. The resulting method—CAT+TempScaling—achieves state-of-the-art performance on most transfer tasks, with on average 7.5% higher accuracy and 4 times lower calibration error compared to domain-adversarial neural networks (DANNs) across the twelve PU transfer tasks.

## 1. Introduction

Fault diagnosis of industrial equipment is a crucial task in prognostics and health management. Intelligent fault diagnosis (IFD) based on deep learning has proven to be an effective and flexible solution, attracting extensive research [1,2]. Deep neural networks are able to learn rich representations from extensive labeled data, allowing them to tackle various tasks across different applications. In IFD in particular, they can achieve high classification performance from sensor data such as time series or spectrograms in an end-to-end manner, without the need for incorporating extensive domain knowledge. However, deep learning models usually only perform well on the data distribution they have been trained on. When applied to a different distribution, they may experience a severe performance drop. This is also observed in fault diagnosis, where the industrial assets are often operated in working conditions different from those in which the labeled data have been collected. Furthermore, obtaining labeled data is difficult and costly in real-world industrial settings. This challenge has recently been addressed recently by deep transfer learning (DTL) and domain adaptation (DA) approaches for IFD [3,4].

### 1.1. Unsupervised Domain Adaptation

Domain adaptation [5] is a type of transfer learning approach aiming at adapting a model from a source domain to a different but related target domain. In particular, unsupervised domain adaptation (UDA) addresses the setting where labeled data are available in the source domain and only unlabeled data are available in the target domain. Domain shift refers to the discrepancies between different domains. In the context of fault diagnosis, domain shift can occur across various scenarios, including different operating conditions [6,7], different units of a fleet [8,9,10,11], data collected under laboratory versus real-world conditions [12], and adaptations from synthetic or simulated data to real data [13,14]. Prevailing approaches for UDA focus on reducing the discrepancy between domains and learning domain-invariant features using the maximum mean discrepancy (MMD) [15], maximum classifier discrepancy (MCD) [16], optimal transport [17], or domain-adversarial training [18]. The latter approach is at the core of domain-adversarial neural networks (DANNs) [18], and has been applied extensively for fault diagnosis [10,11,19].

### 1.2. Self-Training Methods

Self-training [20,21] has emerged as an effective alternative for domain adaptation. First introduced for semi-supervised learning, self-training consists in iteratively generating a set of pseudo-labels [22] on the unlabeled data and retraining the network under the supervision of these pseudo-labels [23,24,25]. However, noisy and inaccurate pseudo-labels hurt the training process. To address this issue, only the most confident predictions are selected for pseudo-labeling, typically using prediction confidence (i.e., the maximum softmax probability) as a proxy for correctness. Curriculum pseudo-labeling (CPL) [26] is a strategy where pseudo-labels are gradually introduced during the learning process in an “easy-to-hard” manner, starting with the most confident target predictions. Once the model is adapted to the target domain, additional samples can be explored. Confident predictions can be selected using a fixed confidence threshold [27] or an adaptive threshold that dynamically adjusts for each class during training to consider the varying difficulties of classes and enable target samples from low-confidence classes to participate early in the training [26,28]. Alternatively, pseudo-labels can be selected by fixing a proportion of the most confident predictions for each class instead of using an explicit threshold value [29,30]. In [31], the variance of predictions between two network sub-branches is used as a replacement for prediction confidence to estimate uncertainty. In their study, French et al. [32] state that the confidence thresholding stabilizes the training and acts as filter to increase the number of correct pseudo-labels.

Two main challenges arise in self-training algorithms [33]: (1) Choosing a trustworthy proxy measure of classification accuracy on unlabeled data; and (2) selecting a threshold on this measure for pseudo-labeling at each training iteration. While solutions have been proposed to tackle the second challenge in the previously discussed literature, in the form of adaptive thresholds capable of handling varying class difficulties during training, the first challenge has remained unaddressed. In this work, we address the first challenge, which is related to uncertainty estimation and model calibration. It is well known that deep neural networks are often badly calibrated and produce over-confident outputs, even for incorrect predictions [34,35,36]. Furthermore, in the presence of a domain shift, the calibration of a model trained on the source domain will degrade even more in the target domain due to the distribution shift [37,38]. In self-training algorithms for domain adaptation, the confidence-based selection of pseudo-labels is hindered by poorly calibrated confidence estimates in the target domain. This limitation restricts the quality of pseudo-labels, leading to error accumulation.

### 1.3. UDA and Self-Training in IFD

In the recent literature, various applications of UDA methods for deep learning-based intelligent fault diagnosis have been explored [3,4]. The main directions explored include domain-adversarial training [39,40,41,42], MMD [43,44], and MCD [45]. Among all possible methods, DANNs are established as a strong and reliable baseline on most benchmark datasets [3]. Despite their effectiveness in aligning source and target features, these methods face challenges with complex distributions and high variability, as they rely on global alignment metrics that may not capture fine-grained class distinctions.

As already mentioned, in recent years, self-training UDA methods based on pseudo-labeling of target samples have emerged. These approaches, which are the main focus of this study, have also been explored for IFD applications. Prediction-consistency-guided convolutional neural networks (PCG-CNNs) [46] draw direct inspiration from [32] and uses a Mean Teacher with a consistency loss, a fixed confidence threshold of 0.96, and a class balance loss. The deep transfer learning with improved pseudo-label learning method (DTL-IPLL) [47] combines MK-MMD feature alignment and pseudo-labeling with a class-wise adaptive threshold as well as a “making-decision-twice” strategy, i.e., predicting twice and discarding the predictions if they differ (which is similar in spirit to Monte Carlo dropout [48]). Wang et al. [49] proposed achieving joint distribution alignment by combining marginal alignment using a DANN with Wasserstein distance and conditional alignment using a triplet loss, using pseudo-labels for target samples. Pseudo-labels are also selected using CPL with a dynamic adaptive threshold. The contrastive cluster center (CCC) [50] approach involves a combination of adversarial training, contrastive cluster alignment and pseudo-labeling, where pseudo-labels of target samples far away from cluster centers are filtered out, but this method does not use self-training. In the semi-supervised learning setting, Ref. [51] uses an aggregation of indicators among which is the entropy on unlabeled samples. Pseudo-labels can also be obtained in an unsupervised way through clustering [52]. Differently, Ref. [53] uses a prototypical network and filters the pseudo-labels using a confidence threshold based on Monte Carlo dropout uncertainty. Finally, Ref. [54] proposes gradually enlarging the set of selected pseudo-labels by using the Euclidean distance in feature space as a measure of confidence, and assigns pseudo-labels using a nearest-neighbor classifier instead of directly predicting with the classifier itself.

However, while these works differ in their use of confidence filtering and distributional alignment techniques, none of them has addressed the challenge of the calibration of confidence estimates for selecting pseudo-labels in the target domain, which may lead to limited robustness and unreliable pseudo-labels, especially in cases of substantial domain shifts where confidence is not well correlated with accuracy in the target domain. Moreover, accurate uncertainty estimates are crucial for trust and adoption by engineers for critical applications, which involves ensuring confidence calibration of models before their deployment.

### 1.4. Contributions

In this paper, we propose a novel UDA method called Calibrated Adaptive Teacher (CAT). The primary novelty of the proposed approach is in improving the calibration in the target domain, with the goal of increasing the accuracy of selected pseudo-labels. CAT consists in a cross-domain teacher–student architecture, where the student network receives confident target pseudo-labels from the teacher network, which is in turn updated by an exponential moving average of the student’s weights. Domain-adversarial feature learning is leveraged to alleviate the domain gap. This architecture is based upon the Adaptive Teacher (AT) [55], recently introduced for object detection in computer vision, and has never been applied to IFD. To address the issue of target-domain calibration, we propose calibrating the predictions of the teacher network on target samples throughout the self-training process, using post hoc calibration techniques such as temperature scaling [34]. Note that previous works on calibration have focused on calibrating already-trained models, which improves the quality of their uncertainty estimates, but has no effect on their performance. However, the novelty of this work relies on the integration of calibration in the self-training procedure in order to improve both the final accuracy and the calibration.

We apply our proposed method to domain-adaptive intelligent fault diagnosis and perform extensive benchmarks and ablation studies on the Paderborn University (PU) bearing dataset, which is characterized by large domain gaps and provides challenging transfer tasks between operating conditions. Experiments are carried out on time-domain and frequency-domain (Fourier transform) inputs, following the benchmark setup by [3]. We demonstrate state-of-the-art performance on most transfer tasks, both with time-domain and frequency-domain inputs. The code is available at https://github.com/EPFL-IMOS/CAT (accessed on 21 November 2024).

The main contributions of this paper are summarized as follows:We propose a novel unsupervised domain adaptation approach, Calibrated Adaptive Teacher (CAT), aiming to improve the calibration of pseudo-labels in the target domain, and thus the overall accuracy on the target data. Our approach consists in introducing post hoc calibration of the teacher predictions during the training.We evaluate our approach on intelligent fault diagnosis and conduct extensive studies on the Paderborn University (PU) bearing dataset, with both time-domain and frequency-domain inputs.CAT significantly outperforms previous approaches in terms of accuracy on most transfer tasks, and effectively reduces the calibration error in the target domain, leading to increased target accuracy.We compare four different post hoc calibration techniques, and demonstrate that temperature scaling [34] and CPCS [37] are the most effective calibration strategies.

The remainder of the paper is organized as follows. In Section 2, we provide the technical background required, before introducing the details of our method in Section 3. Then, we present experimental settings and results in Section 4. Finally, Section 6 concludes this paper.

## 2. Background

### 2.1. Notation

We use the following notation throughout the paper. Since fault diagnostics can be defined as classification problems, we consider *K*-class classification tasks, whereby each class is typically associated with a different fault type. The source (S) and target (T) data are split into training and test sets, with samples and labels, respectively, denoted by $X_{\mathrm{train}}^{S}$, $Y_{\mathrm{train}}^{S}$, $X_{\mathrm{test}}^{S}$, $Y_{\mathrm{test}}^{S}$, and $X_{\mathrm{train}}^{T}$. In the UDA setting, no labels are available in the target domain. The target training labels $Y_{\mathrm{train}}^{T}$ and target test set $X_{\mathrm{test}}^{T}$, $Y_{\mathrm{test}}^{T}$ are used solely for evaluation purposes.

### 2.2. Domain-Adversarial Neural Networks (DANNs)

Domain-adversarial neural networks (DANNs) [18] have emerged as one of the most prominent approaches in UDA. Typically, DANNs comprise a feature encoder, a task-specific module (e.g., a classification head, as in our case), and a domain classifier, also referred to as a discriminator. The feature encoder is shared between the classifier and the discriminator. The underlying principle of this methodology is to train the discriminator to classify the input sample features as belonging to either the source or the target domain. While the discriminator is trained to minimize its classification error, the feature encoder tries to generate indistinguishable features to fool the discriminator, hence the term adversarial. This is commonly achieved using a gradient reversal layer (GRL) [18]. As a result, the marginal distributions of source and target features become aligned, ultimately improving the performance of the source-trained classifier. Domain-adversarial training occurs concurrently with the training of the main task.

### 2.3. Mean and Adaptive Teachers

**Mean Teacher** [56] is a variant of the temporal ensembling [57] approach originally proposed for semi-supervised learning, where knowledge is distilled from a teacher network into a student network. The student is trained with standard gradient updating, whereas the teacher is gradually updated through an exponential moving average (EMA) of the student weights, resulting in an ensemble of all the previous iterations of the student network, increasing its robustness to inaccurate and noisy predictions on unlabeled data. The knowledge distillation can be achieved via a consistency loss between the teacher’s and student’s predicted probabilities, or via hard pseudo-labeling of the unlabeled samples by the teacher with a confidence threshold to increase the number of correct labels from the teacher used in the student’s learning. Mean Teacher was extended to domain adaptation in [32], using a consistency loss on predicted target probabilities and a confidence threshold, as well as a class balance loss to minimize the binary cross-entropy between the mean target class probabilities and a uniform distribution. The confidence threshold is proposed as a replacement to the Gaussian ramp-up used in [56,57], to stabilize the training and act as filter to increase the number of correct pseudo-labels from the teacher used in the student’s learning.

**Adaptive Teacher (AT)** [55] was recently proposed to extend the semi-supervised learning Mean Teacher-based method Unbiased Teacher [58] for UDA in object detection. AT basically combines DANNs and Unbiased Teacher into a single architecture (see Figure 1). Concretely, a domain discriminator is added to the student network to perform feature alignment jointly with mutual teacher–student learning. In AT, the teacher provides hard pseudo-labels of target samples that are filtered using a fixed confidence threshold. Then, the cross-domain student network is trained simultaneously on the labeled source data and the pseudo-labeled target data.

### 2.4. Model Calibration

Classification models typically output conditional probabilities for each class given an input sample $x_{i}$ by applying a softmax activation on the logits: $p_{i}=\sigma \left(z_{i}\right)$. The predicted class is ${\hat{y}}_{i}={\arg\max}_{y}p_{i}\left[y\right]$, and its corresponding probability is called the prediction confidence:(1)$$c_{i}:=\max p_{i}=p_{i}\left[{\hat{y}}_{i}\right].$$

A classifier is said to be well calibrated if the confidence estimates are equal to the true accuracy of the predictions. For instance, if the confidence is equal to 0.9, the prediction should be correct 90 percent of the time. The calibration of a model can be represented visually by a reliability diagram [34,59], plotting the expected accuracy as a function of average confidence (see, e.g., Figure 3).In order to estimate these quantities from a finite dataset, we divide the [0,1] interval into *M* equally spaced bins of size 1/M, and partition the samples into groups $B_{m}:=\left\{i\;;\;\frac{m-1}{M}<c_{i}\leq \frac{m}{M},1\leq m\leq M\right\}$. Then, we define the expected accuracy and average confidence of a bin as follows:(2)$$\begin{array}{cc} \; & \mathrm{acc}\left(B_{m}\right):=\frac{1}{|B_{m}|}\sum_{i\in B_{m}}1\left({\hat{y}}_{i}=y_{i}\right) \end{array}$$(3)$$\begin{array}{cc} \; & \mathrm{conf}\left(B_{m}\right):=\frac{1}{|B_{m}|}\sum_{i\in B_{m}}c_{i}. \end{array}$$
where $y_{i}$ is the true label and 1 is the indicator function, equal to 1 when its argument is true and 0 otherwise. In a perfectly calibrated model, these quantities are supposed to be equal. Whenever we have $\mathrm{acc}\left(B_{m}\right)<\mathrm{conf}\left(B_{m}\right)$, the model is said to be over-confident, and in the opposite case, it is said to be under-confident. Hence, a well-calibrated model should produce a reliability diagram close to the diagonal. To summarize the calibration quality, a commonly used metric is the expected calibration error (ECE) [60]:(4)$$\mathrm{ECE}:=\sum_{m=1}^{M}\frac{|B_{m}|}{N}\left|\mathrm{acc}\left(B_{m}\right)-\mathrm{conf}\left(B_{m}\right)\right|$$
where *N* is the total number of samples. Its value is in the [0,1] range, with 0 corresponding to a perfectly calibrated model. We use a number of bins of M=10 in the ECE computations throughout the paper, following [60].

While a well-calibrated model is desirable, it was shown empirically that neural networks are often badly calibrated in classification [34], regression [35], and anomaly detection [36]. In response, several techniques for network calibration have been developed, which can be broadly classified into post hoc, train time, and through out-of-distribution detection [61]. Additionally, it was also demonstrated that calibration degrades under domain shift [62]. Thus, in the setting of UDA, models are poorly calibrated in the target domain [38].

## 3. Proposed Method

### 3.1. Overview

We propose the Calibrated Adaptive Teacher (CAT) framework for unsupervised domain adaptation, an extension of the Adaptive Teacher (AT) [55]. CAT aims to improve the quality of pseudo-labels generated by the teacher network in the target domain. The primary innovation of this architecture lies in introducing post hoc calibration into the self-training process.

The architecture, summarized in Figure 1, comprises two networks: a teacher and a student network, sharing the same architecture but with different weights.

**Student (see Figure 1, on green background).** In the student network, both source and target inputs first pass through a feature encoder (➀). This encoder consists of a 1D-CNN followed by a 256-dimensional bottleneck, as detailed in [3]. Subsequently, source and target fault classes are predicted by a fully connected (FC) linear layer with softmax activation (➁). The source and target features are also fed into the gradient reversal layer (GRL) (➂) and domain classifier (DC) for domain-adversarial training (➃). The weights of each component in the student network are then updated through gradient descent training, using the loss function detailed in the following paragraphs.

**Teacher (see Figure 1, on orange background).** On the teacher side, only target inputs go through a feature encoder (➀) and a linear classification head (➁). Teacher predictions are then used to supervise the student through pseudo-labeling. Before selecting the pseudo-labels based on confidence, we introduce post hoc calibration (➄) to calibrate the teacher predictions, which is the primary novelty in this architecture. Essentially, this calibration transforms the target logits (i.e., the outputs before the softmax) to obtain better-calibrated probabilities after applying the softmax. We then select the most confident predictions as pseudo-labels, using a class-wise adaptive threshold instead of the fixed threshold used in [55] (➅). These pseudo-labels serve as supervision for the student network to train on target samples. The teacher weights are frozen and updated through the EMA of the student weights between each training iteration, as presented in Figure 1.

We compare existing UDA approaches for IFD based on self-training with pseudo-labels in Table 1. To the best of our knowledge, none of these works has investigated the aspect of model calibration. The details of each layer in the architecture are provided in Table 2. Following this overview of our proposed approach, we introduce the key components in more detail.

### 3.2. Calibrated Self-Training

A model is considered well calibrated if its confidence scores align with the accuracy of predictions. Since our objective is to increase the accuracy of the selected pseudo-labels, and confidence serves as a proxy for accuracy, we aim for well-calibrated target outputs. Therefore, we propose calibrating the teacher network’s outputs before selecting confident pseudo-labels for training. To achieve this calibration, we leverage a post hoc calibration technique. Different post hoc calibration techniques are available for multi-class classification, with differences in effectiveness and the number of parameters. In this study, we compare four previously introduced post hoc calibration techniques: temperature scaling, vector scaling, matrix scaling, and calibrated predictions with covariate shift (CPCS) [37]. All these techniques rely on a labeled hold-out test set from the source domain to learn a transformation of the target logits. Temperature scaling, with a single scalar parameter, was identified to be the most effective method in the study by Guo et al. [34]. Additionally, we evaluate vector and matrix scaling, which adjust each class differently, requiring 2K and $K^{2}+K$ parameters, respectively, where *K* is the number of classes. The first three techniques do not account for distribution shifts. In the context of a domain shift, we also evaluate CPCS, which addresses covariate shift by applying importance weighting with a logistic domain discriminator on domain-invariant features. These features are obtained through the domain classifier in our approach.

**Definition** **1**(Calibrated self-training)**.** *Let x be an unlabeled input and z be the logits obtained from the teacher network before the softmax activation σ. The calibrated teacher predictions are defined as*
(5)$$p_{\mathrm{teacher}}^{\mathrm{cal}}\left(x\right)=\sigma \left(f\left(z\right)\right)$$
*where $f:R^{K}\to R^{K}$ is a calibration function operating on the logits and K is the number of classes.*

We propose four variants of CAT, each incorporating different post hoc calibration techniques for multi-class classification.

**CAT-TempScaling** Temperature scaling is a common approach that involves scaling the output logits using a single scalar parameter *T*, called temperature. This parameter is tuned to minimize the negative log-likelihood on a hold-out test set in the source domain. The newly calibrated probabilities are expressed as $p_{i}=\sigma \left(z_{i}/T\right)$. The calibration function is defined as follows:(6)$$\begin{array}{cc} \; & f:z\mapsto z/T^{★}\\ & \;\mathrm{where}\;T^{★}=\underset{T\in R}{\arg\min}-\frac{1}{|X_{\mathrm{test}}^{S}|}\sum_{\left(x,y\right)}\sum_{k=1}^{K}1\left(y=k\right)\cdot \log\sigma \left(z/T\right)\left[k\right] \end{array}$$
where the first summation is over the source test samples and labels $X_{\mathrm{test}}^{S}\times Y_{\mathrm{test}}^{S}$. **CAT-VectorScaling** Vector scaling involve transforming the logits using a linear transformation with a matrix W and a bias vector b: $p_{i}=\sigma \left(Wz_{i}+b\right)$, where the matrix is restricted to be diagonal, and the parameters W,b are tuned to minimize the negative log-likelihood on a hold-out source test set. The calibration function is defined as follows:(7)$$\begin{array}{cc} \; & f:z\mapsto W^{★}z+b^{★}\\ & \;\mathrm{where}\;W^{★},b^{★}=\underset{\left(W,b\right)\in \mathrm{diag}\left(K\right)\times R^{K}}{\arg\min}-\frac{1}{|X_{\mathrm{test}}^{S}|}\sum_{\left(x,y\right)}\sum_{k=1}^{K}1\left(y=k\right)\cdot \log\sigma \left(Wz+b\right)\left[k\right] \end{array}$$

**CAT-MatrixScaling** Matrix scaling scales the logits using a similar transformation, but where the matrix can be non-diagonal. The corresponding calibration function is defined as follows:(8)$$\begin{array}{cc} \; & f:z\mapsto W^{★}z+b^{★}\\ \; & \;\mathrm{where}\;W^{★},b^{★}=\underset{\left(W,b\right)\in R^{K\times K}\times R^{K}}{\arg\min}-\frac{1}{|X_{\mathrm{test}}^{S}|}\sum_{\left(x,y\right)}\sum_{k=1}^{K}1\left(y=k\right)\cdot \log\sigma \left(Wz+b\right)\left[k\right] \end{array}$$

**CAT-CPCS** Calibrated prediction with covariate shift (CPCS) [37] proposes to correct for covariate shift by combining adversarial feature alignment (as in DANNs [18]), importance weighting using a logistic domain discriminator, and temperature scaling. First, domain-invariant features are learned through domain-adversarial training. Then, a logistic domain discriminator is trained to distinguish between the features extracted from $X_{\mathrm{train}}^{S}$ and $X_{\mathrm{train}}^{T}$, and its output probabilities are used to estimate the importance weight w(x) for each sample. Specifically, the weight is equal to the ratio between the probability of belonging to the target domain and the probability of belonging to the source domain, allowing for a translation from the source distribution to the target one. Finally, the optimal temperature is found by minimizing the weighted Brier score [63] (i.e., the mean squared error between outputs and one-hot labels) on the hold-out source test set. The calibration function is defined as follows:(9)$$\begin{array}{cc} \; & f:z\mapsto z/T^{★}\\ \; & \;\mathrm{where}\;T^{★}=\underset{T\in R}{\arg\min}\frac{1}{|X_{\mathrm{test}}^{S}|}\sum_{\left(x,y\right)}w\left(x\right)\sum_{k=1}^{K}{\left(1(y=k)-\sigma (z/T)[k]\right)}^{2} \end{array}$$

All optimization problems for finding the optimal calibration parameters are convex and solved using the optimize.fmin function from scipy.

Note that this step adds computational overhead. Nevertheless, as we update the calibration function only once per epoch, the added cost is minimal. Moreover, the computational complexity of the post hoc calibration is typically relatively low. In TempScaling, the complexity is driven by the computation of the log-likelihood, which is linear in both the test set size and the number of classes, and the number of iterations of the temperature optimization, which is very small for tuning a single scalar parameter and considered constant. Vector and MatrixScaling tune, respectively, 2K and $K^{2}+K$ parameters, adding only a slightly higher constant term due to the higher dimension. CPCS requires, in addition, to train a logistic domain discriminator to compute the importance weights, adding up to a cost with linear complexity in the number of samples and iterations (gradient-based training). No additional adversarial feature alignment is required, as CAT already learns aligned features thanks to the domain classifier.

### 3.3. Adaptive Confidence Threshold

Curriculum pseudo-labeling methods dynamically adjust the confidence threshold for each class during training, based on the accuracy of each class. We adopt the method introduced in [26]. At training iteration *t*, the dynamic threshold for class *k* is defined as
(10)$$T_{t}\left(k\right)=a_{t}\left(k\right)\cdot \tau$$
where $a_{t}\left(k\right)$ represents the accuracy, and τ is a fixed threshold value. As proposed in [26], the accuracy can be substituted with the *learning effect* of the class, which is reflected by the number of high-confidence predictions for this class:(11)$$\sigma_{t}\left(k\right)=\sum_{x\in X_{\mathrm{train}}^{T}}1\left(\max p_{\mathrm{teacher}}\left(x\right)\geq \tau \right)\cdot 1\left({\arg\max}_{y}p_{\mathrm{teacher}}\left(x\right)\left[y\right]=k\right)$$

Subsequently, this quantity is scaled, undergoes a non-linear mapping M, and is ultimately used to define the dynamic threshold as follows:(12)$$\begin{array}{cc} \beta_{t}\left(k\right) & =\frac{\sigma_{t}\left(k\right)}{\max_{k}\sigma_{t}\left(k\right)}\\ M(x) & =\frac{x}{2-x}\\ T_{t}\left(k\right) & =M(\beta_{t}\left(k\right))\cdot \tau \end{array}$$

At training iteration *t*, we ultimately define the set of selected target pseudo-labels:(13)$$\begin{array}{cc} \; & {\hat{X}}_{\mathrm{train}}^{T}=\left\{x\;;\;x\in X_{\mathrm{train}}^{T},\max p_{\mathrm{teacher}}\left(x\right)\geq T_{t}\left({\arg\max}_{y}p_{\mathrm{teacher}}\left(x\right)\left[y\right]\right)\right\} \end{array}$$(14)$$\begin{array}{cc} \; & {\hat{Y}}_{\mathrm{train}}^{T}=\left\{{\arg\max}_{y}p_{\mathrm{teacher}}\left(x\right)\left[y\right]\;;\;x\in X_{\mathrm{train}}^{T},\max p_{\mathrm{teacher}}\left(x\right)\geq T_{t}\left({\arg\max}_{y}p_{\mathrm{teacher}}\left(x\right)\left[y\right]\right)\right\} \end{array}$$

In our CAT method, the teacher probabilities $p_{\mathrm{teacher}}\left(x\right)$ are replaced with the calibrated teacher probabilities $p_{\mathrm{teacher}}^{\mathrm{cal}}\left(x\right)$.

### 3.4. Loss Function and Training Procedure

#### 3.4.1. Student Training

We denote the parameters of the student network as $\theta =\{W_{E},W_{C}\}$, where $W_{E}$ and $W_{C}$ represent the weights of the student encoder and classification head, respectively. The weights of the discriminator are denoted as $W_{\mathrm{DC}}$. Instead of the standard DANN [18], we adopt the enhanced approach of smooth domain-adversarial training (SDAT) [64]. This involves using the Sharpness Aware Minimization (SAM) optimizer [65] on the task loss (i.e., supervised source loss) and incorporating the minimum class confusion (MCC) loss [66], which stands as a current state-of-the-art method for domain adaptation in computer vision tasks. The MCC loss serves as a non-adversarial regularization term designed to minimize pairwise class confusion in the target domain.

The loss function of CAT, denoted $L_{\mathrm{CAT}}$, comprises four terms: a supervised source loss $L_{C}$, a target pseudo-labeling loss $L_{\mathrm{PL}}$, the domain classifier term $L_{\mathrm{DC}}$, and the MCC loss $L_{\mathrm{MCC}}$. The supervised loss is calculated as the cross-entropy between source predictions and labels:(15)$$L_{C}\left(\theta ;X_{\mathrm{train}}^{S},Y_{\mathrm{train}}^{S}\right)=-\frac{1}{|X_{\mathrm{train}}^{S}|}\sum_{\left(x,y\right)}\sum_{k=1}^{K}1\left(y=k\right)\cdot \log p\left(x\right)\left[k\right]$$
where the first summation is over the source training samples and labels $X_{\mathrm{train}}^{S}\times Y_{\mathrm{train}}^{S}$. The pseudo-labeling loss is then calculated as the cross-entropy between predictions and the pseudo-labels:(16)$$L_{\mathrm{PL}}\left(\theta ;{\hat{X}}_{\mathrm{train}}^{T},{\hat{Y}}_{\mathrm{train}}^{T}\right)=-\frac{1}{|{\hat{X}}_{\mathrm{train}}^{T}|}\sum_{\left(x,y\right)}\sum_{k=1}^{K}1\left(y=k\right)\cdot \log p\left(x\right)\left[k\right]$$
where the first summation is over the pseudo-labels ${\hat{X}}_{\mathrm{train}}^{T}\times {\hat{Y}}_{\mathrm{train}}^{T}$. The domain classifier loss is then expressed as a binary cross-entropy between the domain classifier predictions and domain labels, which is set to 1 for the source and 0 for the target training samples:(17)$$L_{\mathrm{DC}}\left(W_{E},W_{\mathrm{DC}};X_{\mathrm{train}}^{S},X_{\mathrm{train}}^{T}\right)=-\frac{1}{|X_{\mathrm{train}}^{S}|}\sum_{x\in X_{\mathrm{train}}^{S}}\log p_{\mathrm{DC}}\left(x\right)-\frac{1}{|X_{\mathrm{train}}^{T}|}\sum_{x\in X_{\mathrm{train}}^{T}}\log\left(1-p_{\mathrm{DC}}\left(x\right)\right)$$

Finally, the total loss is an equally weighted sum of the individual loss terms:(18)$$L_{\mathrm{CAT}}\left(\theta ,W_{\mathrm{DC}};X_{\mathrm{train}}^{S},Y_{\mathrm{train}}^{S},{\hat{X}}_{\mathrm{train}}^{T},{\hat{Y}}_{\mathrm{train}}^{T}\right)=L_{C}-L_{\mathrm{DC}}+L_{\mathrm{PL}}+L_{\mathrm{MCC}},$$
omitting the arguments of each loss term for brevity. Source and target cross-entropies are given equal weights, and Ref. [66] found that a weight of 1 for the MCC loss worked across all their experiments. For the domain-adversarial training, we adopt the same strategy as [18], only used in [3], progressively increasing the GRL coefficient from 0 to 1 following the formula $\frac{2}{1+\exp\left(-10p\right)}$, where *p* increases linearly from 0 to 1 during training. The total loss is minimized with respect to the student weights, while the domain classifier loss is minimized with respect to the discriminator weights and maximized with respect to the encoder weights in an adversarial way via the gradient reversal layer. Thus, the overall objective is expressed as follows:(19)$$\min_{\theta }\;\max_{W_{\mathrm{DC}}}\;L_{\mathrm{CAT}}\left(\theta ,W_{\mathrm{DC}};X_{\mathrm{train}}^{S},Y_{\mathrm{train}}^{S},{\hat{X}}_{\mathrm{train}}^{T},{\hat{Y}}_{\mathrm{train}}^{T}\right)$$

#### 3.4.2. Teacher Updating

The teacher network parameters are initialized with the student weights at the beginning of self-training. Subsequently, they are updated between each training iteration by the EMA of the student weights with an update rate α:(20)$$\theta_{\mathrm{teacher}}\leftarrow \alpha \cdot \theta_{\mathrm{teacher}}+\left(1-\alpha \right)\cdot \theta$$

### 3.5. Training Procedure of the Entire Pipeline

The CAT training procedure consists of three phases. The first phase involves a source-only training of the student network using only the supervised loss $L_{C}$. In the second phase, the domain classifier loss is enabled after $T_{\mathrm{DA}}$ epochs for domain-adversarial training. The third phase, mutual teacher–student training, begins at $T_{\mathrm{PL}}$, with $T_{\mathrm{DA}}\leq T_{\mathrm{PL}}$. A warm-up phase is necessary to ensure sufficient pseudo-label quality and to avoid compromising the training by fitting noise [30]. Lastly, the calibration is enabled later in the training at $T_{\mathrm{cal}}$, with $T_{\mathrm{PL}}\leq T_{\mathrm{cal}}$.

During the mutual training phase, at the beginning of each given epoch *t*, we estimate the class-wise adaptive threshold values $T_{t}\left(k\right)$ following (Equation 12), as well as the calibration function following (Equation 6) in CAT-TempScaling, (Equation 7) in CAT-VectorScaling, (Equation 8) in CAT-MatrixScaling, and (Equation 9) in CAT-CPCS, depending on the variant of CAT considered. These estimations involve the unlabeled target training set and the labeled source test set. At every training iteration, a batch of inputs and labels is sampled from the source training set, and an unlabeled batch of inputs is sampled from the target training set. We compute and calibrate the logits of the teacher network to obtain calibrated probabilities, which are subsequently filtered using the adaptive threshold following (Equation 13,14). We then minimize the objective (Equation 19) by taking a gradient descent step. Finally, the teacher weights are updated by EMA following (Equation 20).

The complete training procedure is detailed in Algorithm 1.   
**Algorithm 1:** Calibrated Adaptive Teacher (CAT) training procedure
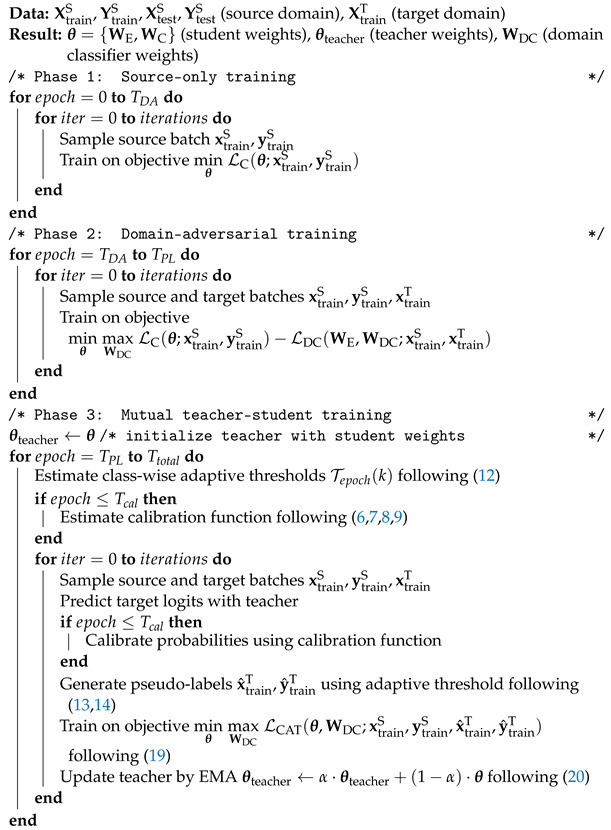


## 4. Experiments

In this section, we discuss the data and transfer learning tasks, and provide details on the experimental settings and hyperparameters.

### 4.1. Case Study

Experiments were conducted on the bearing fault diagnostics Paderborn University (PU) dataset [67], which comprises challenging transfer learning tasks across various operating conditions and stands as a benchmark in the recent literature [3,68,69,70]. The dataset comprises motor current signals of an electro-mechanical drive system, allowing for bearing diagnostics without the need for additional acceleration sensors for vibration analysis. The domain adaptation tasks involve adapting between four different operating conditions—rotational speed, load torque and radial force—described in Table 3.

From a domain knowledge and signal processing point of view, domain shifts of varying intensities can be expected. Varying the speed, torque, or force has very different impacts on bearing vibration signals characteristics [71]. Thus, we can expect each domain shift to result in various transfer task difficulties. Changing the speed (transfer tasks 0↔1, 1↔2 and 1↔3) significantly impacts both the frequency content—frequencies will be scaled due to rotational speed, and it can introduce harmonics, broadening the spectrum—and the amplitude. Changing the force (transfer tasks 0↔3, 1↔3 and 2↔3) increases the mechanical stress in the system, which also significantly affects the amplitude, as well as the frequency content to a lesser extent. Torque (transfer tasks 0↔2, 1↔2 and 2↔3) has less impact on the signals, resulting in easier tasks, especially when other parameters are left fixed (tasks 0↔2).

We adopt the same setting as [3] for a fair comparison. The classification problem has 13 classes and consists in classifying different combination of fault types and severities, represented by different bearing codes, described in Table 4. Each sample consists of a 1024-dimensional input in the time-domain, or in the frequency-domain after fast Fourier transform.

### 4.2. Training Parameters

We train the model for a total of $T_{\mathrm{total}}=300$ epochs with a batch size of 64, using the Adam optimizer with a learning rate of 0.001, $\beta_{1}=0.9$, $\beta_{2}=0.999$, and an $\ell_{2}$ weight decay of $10^{-5}$. The learning rate is reduced by a factor of 10 at epochs 150 and 250. The domain-adversarial loss is introduced at epoch $T_{\mathrm{DA}}=50$. Self-training starts at epoch $T_{\mathrm{PL}}=50$ for the PU time domain, and at epoch $T_{\mathrm{DA}}=100$ for the PU frequency domain. In both cases, we introduce the calibration from epoch $T_{\mathrm{cal}}=150$. The teacher EMA update rate is set to 0.999. The fixed threshold value in the adaptive threshold is τ=0.9. No weak–strong data augmentations were used. The train–test data splitting follows [3], with 80% of total samples for training and 20% for testing.

## 5. Results

In this section, we present the results of experiments on the PU dataset. First, we demonstrate the impact of our proposed calibration method on the quality of pseudo-labels and model performance in Section 5.1. Following this, Section 5.2 and Section 5.3 present a quantitative benchmark analysis comparing different UDA methods in terms of performance and calibration. We conduct ablation studies in Section 5.4.

### 5.1. Calibration and Quality of Pseudo-Labels

We begin by illustrating the motivation behind our approach, which is to address the issue of model calibration in the target domain, aiming to increase the accuracy of pseudo-labels.

In Figure 2, we present the reliability diagrams of the teacher predictions of a trained AT model on the source and target test sets. Blue bars under the diagonal represent over-confident predictions and vice-versa. The red area visualizes the gap between actual and ideal calibration. The initial observation is that the model is over-confident in both domains, but the expected calibration error (ECE) is even higher in the target domain (36.90%) than in the source domain (21.27%).

We present the same reliability diagrams after applying the temperature scaling (i.e., CAT-TempScaling) in Figure 3. Calibration drastically reduces the calibration error on the source domain to 5.41%, as expected, since the temperature was tuned on the source test set. However, it also significantly reduces the ECE on the target domain (down to 10.56%), even though temperature scaling does not account for domain shift. Thanks to feature alignment, the source and target distributions are similar enough to allow for a transfer of temperature scaling to the target data.

As a consequence, we can expect the quality of the selected pseudo-labels to improve, as the confidence more closely matches the true accuracy, and we reduce the number of over-confident wrong target predictions. Figure 4 displays the evolution of target pseudo-label accuracy during training for the transfer task 0→1, comparing AT + MCC + SDAT (denoted by AT*) and our proposed methods CAT*-TempScaling and CAT*-CPCS. At the epoch where calibration is introduced, we observe an absolute increase in pseudo-label accuracy of around 10%, which is maintained during the rest of the training. At the last iteration, pseudo-label accuracy reaches 47.64% for AT*, and 57.37% and 58.45% for CAT*-TempScaling and CAT*-CPCS, respectively.

In addition, Figure 5 illustrates the evolution of the target test accuracy and ECE during training for the different methods. Specifically, we compare the source-only model, DANN, DANN + MCC + SDAT (denoted as DANN*), AT + MCC + SDAT (denoted as AT*), and our proposed methods CAT*-TempScaling and CAT*-CPCS. Domain adaptation and self-training both start at $T_{\mathrm{DA}}=T_{\mathrm{PL}}=50$ epochs, and calibration is enabled at $T_{\mathrm{cal}}=150$ epochs. All domain adaptation methods drastically improve the performance compared to the source-only model. AT significantly improves over the DANN baseline, but still exhibits a high ECE. After introducing calibration, the ECE drops significantly (see Figure 5, right), and the increased pseudo-label accuracy also translates into an improvement in accuracy on the target validation set (see Figure 5, left).

Although TempScaling and CPCS effectively improve calibration in the target domain, this is not always the case for VectorScaling and MatrixScaling. The reliability diagrams shown in Figure 6 and Figure 7 demonstrate that these two calibration techniques are able to reduce calibration error more effectively than temperature scaling in the source domain. However, in the target domain, calibration error is not reduced. This observation shows that the higher number of parameters (scaling matrix and bias vector) of these calibration techniques prevents them from generalizing to the target domain, as opposed to temperature scaling and CPCS, which only tune a single scalar parameter on the source test set. In other words, the scaling parameters are too specific to the source domain.

### 5.2. Comparative Analysis of Performance

In this section, we conduct a comparative analysis on the 12 transfer tasks of the PU bearing fault diagnosis benchmark, considering both time-domain and frequency-domain (FFT) inputs. In most related works, only time-domain inputs are considered (see Table 1). The compared methods include the model trained on the source (source-only), DANN, which is the best-performing domain adaptation method overall in the survey [3] and represents our main baseline, and DANN + MCC + SDAT (denoted as DANN*, where the asterisk indicates that the model integrates the MCC loss and SDAT [64]). We also evaluate AT*, our adaptation of AT [55]. Finally, we assess our proposed calibrated self-training methods CAT* with four different post hoc calibration techniques, namely, CAT*-TempScaling, CAT*-CPCS, CAT*-VectorScaling and CAT*-MatrixScaling. In the case of teacher–student models, we always report the results of the teacher network throughout the paper.

We use the same parameters as [3] for the source-only and DANN training. We assess the performance at the last training iteration, as using the test labels for early stopping is unrealistic. Thus, we compare our results to the “Last-Mean” results of [3]. All our experiments are repeated five times with different random seeds. As performance measures, we report the average accuracy per transfer task and the overall average accuracy. Additionally, we report the average rank, which is better suited for comparing multiple methods over multiple tasks of varying difficulties [72].

The results for the PU dataset in the time domain are presented in Figure 8 and Table 5. The domain adaptation baseline DANN achieves an overall accuracy of 46.53% across all tasks, consistent with the results reported by [3], and significantly higher than the source-only model (33.78%). It is worth noting that some transfer tasks are easy, with the source model already performing well (0→2 and 2→0), while other tasks are challenging, with accuracies below 50%. DANN* (DANN + MCC + SDAT) has a slightly higher average accuracy of 47.02%. The AT* method obtains a significantly higher average accuracy of 53.13%, outperforming DANN* on every task except the easiest one (0→2). Overall, the best-performing methods are our proposed CAT*-TempScaling and CAT*-CPCS, with average accuracies of 54.50% and 54.42%, and average ranks of 2.28 and 2.40. The calibrated methods outperform the non-calibrated versions in 10 out of 12 tasks. Among the four calibration techniques, we observe that only TempScaling and CPCS are effective, whereas VectorScaling and MatrixScaling degrade the performance in most tasks.

The results for the frequency-domain inputs are presented in Figure 9 and Table 6. The findings are consistent with those in the time domain, and the methods exhibit similar average ranks. The source-only performance is higher than in the time domain, achieving 44.04% accuracy. The improvement of DANN* (59.84%) over DANN (57.89%) is more pronounced. Once again, the AT* method demonstrates a high average accuracy (63.94%) and significantly outperforms DANN* on all the challenging tasks, while showing comparable performance on the easier ones. As in the time domain, our proposed CAT*-TempScaling and CAT*-CPCS achieve the best performance, achieving 64.76% and 64.92% accuracy, with an equal average rank of 2.76 out of the eight compared methods. CAT*-VectorScaling and CAT*-MatrixScaling, however, fail to improve upon AT*.

To assess the statistical significance of the differences between compared methods, we conducted a statistical post hoc analysis using pairwise Wilcoxon signed-rank tests with Holm’s step-down procedure at a significance level of 0.05 [72]. This procedure compares the sums of ranks across the 12 transfer tasks and five runs, testing against the null hypothesis stating that methods have equal performance. In Table 5, Table 6 and Table 7, the average ranks are reported in the rightmost column, with the rank of the best method and statistically equivalent methods in bold. As a result, our proposed CAT*-TempScaling and CAT*-CPCS have higher ranks than AT*, and this difference is significant in both the time and frequency domains. However, both calibration techniques are equivalent, indicating that the importance weighting in CPCS is not necessary in this case. The features are sufficiently well aligned to make temperature scaling directly transferable and effective.

The result of the statistical tests are summarized in critical difference diagrams [72], shown in Figure 10. These diagrams visualize the average ranks of each method in terms of target accuracy, and connect statistically equivalent methods with a horizontal bar (i.e., when the null hypothesis could not be rejected at a significance level of 0.05). In both the time and frequency domains, CAT*-TempScaling and CAT*-CPCS emerge as the top-performing methods, with equivalent performance. The next best methods, AT* and CAT*-VectorScaling, have significantly different ranks with time-series inputs but not with FFT inputs. The CD-diagram (https://github.com/hfawaz/cd-diagram accessed on 13 October 2024) library was used to create these diagrams.

### 5.3. Comparative Analysis of Calibration Error

In this section, we compare the calibration error, in terms of expected calibration error (ECE), across different methods. The ECE of the target-domain predictions is presented in Figure 11 and detailed in Table 7 for time-domain inputs, with frequency-domain results available in Appendix A. On average, both the source-only model and DANN exhibit very high ECE values, exceeding 50% on most tasks, particularly the challenging ones. While the Adaptive Teacher shows improved calibration, it still maintains a relatively high average ECE of 34.26%. CAT*-VectorScaling and CAT*-MatrixScaling fail to improve calibration in the target domain on most tasks compared with AT*, aligning with their subpar accuracy performance. As observed previously, their higher number of parameters (scaling matrix and bias vector) prevent them from generalizing to the target domain due to this over-parametrization, as opposed to temperature scaling and CPCS, which only tune a single scalar parameter on the source test set. Finally, CAT*-TempScaling and CAT*-CPCS achieve the lowest calibration error, with average ECE values of 12.77% and 11.31%, and average ranks of 1.67 and 1.48, respectively.

### 5.4. Ablation Studies

In this section, we present the results of ablation studies, providing justification for the design choices in our proposed method and assessing the impact of each component. Initially, we compare the performance of the AT model using a fixed confidence threshold (τ=0.9) against an adaptive threshold, as introduced in Section 3. The results on the PU dataset in the time domain (see Table 8) indicate that, in most transfer tasks, particularly the most challenging ones, the adaptive threshold outperforms the fixed threshold in terms of target-domain teacher accuracy. The adaptive thresholding strategy achieves an average accuracy of 48.33%, whereas the fixed thresholding strategy averages 46.22% over all tasks.

In a second ablation study, we investigated the influence of incorporating the MCC loss [66] and the SDAT optimization technique [64] into the AT and CAT methods. We conducted experiments for AT, CAT-TempScaling and CAT-CPCS on the PU dataset in time domain, using different combinations of MCC and/or SDAT, as reported in Table 9. For each method, using MCC and SDAT together results in a substantial performance gain, consistent with findings by [64]. The average target-domain accuracy of the teacher network across all tasks is approximately 4% higher when using both MCC and SDAT compared to not using them.

## 6. Conclusions

In this paper, we tackled the challenge of model calibration for pseudo-labeling in the context of unsupervised domain adaptation. We proposed a novel method called Calibrated Adaptive Teacher (CAT), drawing inspiration from Mean Teacher, self-training with pseudo-labels, and feature alignment through domain-adversarial training. The primary innovation involves calibrating the predictions of the teacher network in the target domain throughout the self-training process. We explored four post hoc calibration techniques, with temperature scaling and CPCS yielding the best results. Interestingly, both techniques performed similarly, despite the fact that the first one does not account for the domain shift. We believe this is due to the presence of already well-aligned features when calibration is introduced, enabling temperature scaling to transfer to the target domain. Experiments on intelligent fault diagnosis demonstrated that our method is able to improve calibration in the target domain, resulting in increased accuracy. On the Paderborn University bearing dataset, our method outperformed previous unsupervised domain adaptation approaches by a significant margin: on average, with time-series inputs, accuracy is 7.5% higher and calibration error is 4 times lower than DANN across the twelve PU transfer tasks. With frequency-domain inputs, the improvement is +5% in accuracy and there is a 2 times lower calibration error.

Our method is effective whenever the model is badly calibrated in the target domain, which is often the case in deep learning. The main limitation of our approach is that if target predictions are already well calibrated, improvement cannot be expected. Additionally, we observed that our approach is more effective in tackling more challenging tasks, specifically when the baseline accuracy is relatively low, without adaptation, and predictions are poorly calibrated. For example, in the transfer tasks 0→2 and 2→0, the source model already shows high accuracy (over 75%, while harder tasks reach under 30%) and relatively low ECE (around 18%, while other tasks are over 50%), and CAT produces no significant performance gain on these tasks compared with DANN.

The application of weak–strong data augmentations, as implemented in [21,55], requires further investigation in the context of intelligent fault diagnosis. The use of weak augmentations for the teacher network and strong augmentations for the student network is highly beneficial in computer vision applications. Exploring suitable augmentations for time-series and spectrum data represents a promising direction for future research.

## Figures and Tables

**Figure 1 sensors-24-07539-f001:**
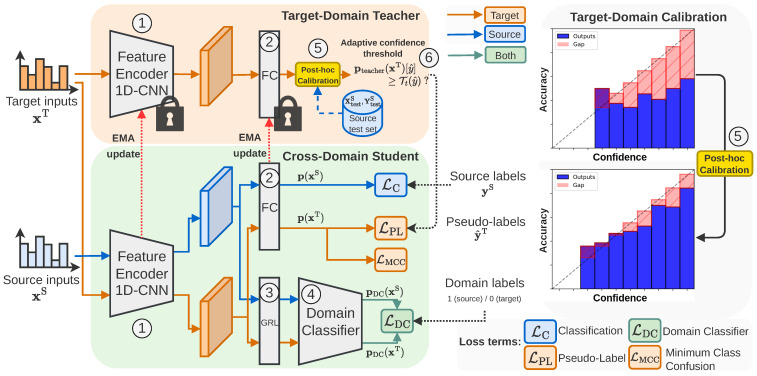
Our proposed Calibrated Adaptive Teacher (CAT). The main novelty involves a post hoc calibration of the teacher predictions in the target domain throughout the self-training process, improving the quality of pseudo-labels.

**Figure 2 sensors-24-07539-f002:**
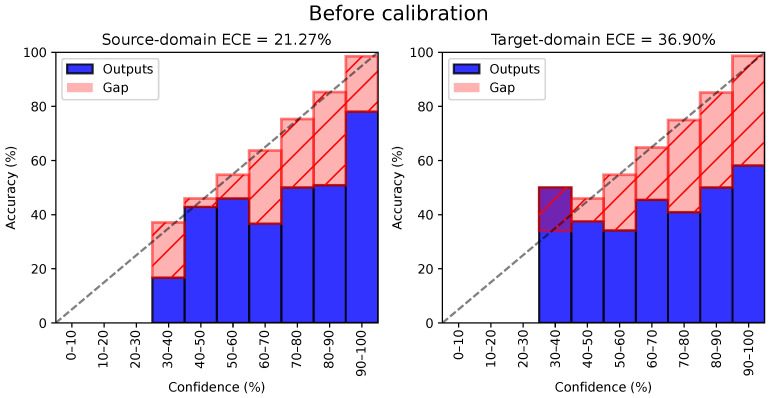
Example of reliability diagram before applying calibration (here, AT on the 0→1 task in time domain). The model has higher expected calibration error (ECE) on the target domain than on the source domain. The dotted line represents ideal calibration.

**Figure 3 sensors-24-07539-f003:**
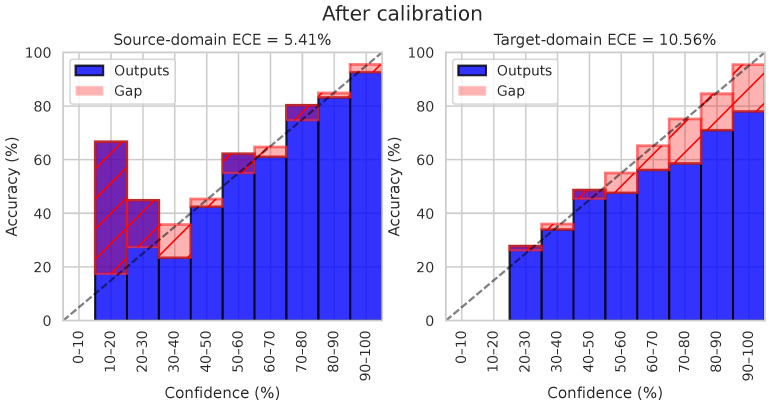
Example of of reliability diagram after applying temperature scaling (here, CAT on the 0→1 task in time domain). Even though temperature scaling is based on the source validation set, ECE is also drastically reduced on the target domain, owing to well-aligned features. The dotted line represents ideal calibration.

**Figure 4 sensors-24-07539-f004:**
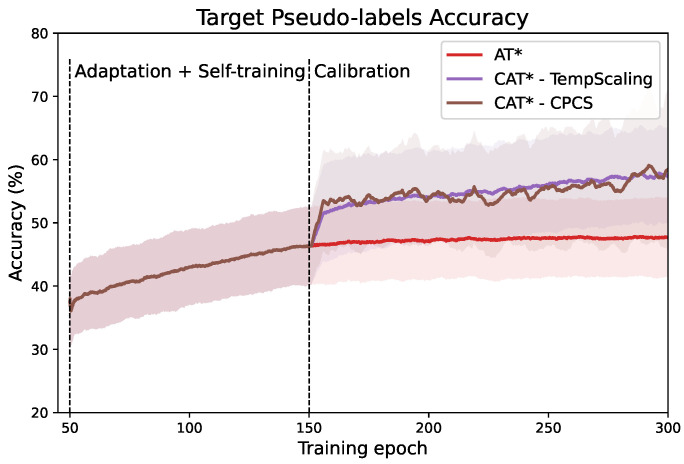
Evolution of target pseudo-label accuracy produced by the teacher network during training for different methods. A boost in accuracy is observed after introduction of the calibration in our proposed CAT. Mean ± standard deviation over 5 runs with different random seeds. * = with MCC + SDAT.

**Figure 5 sensors-24-07539-f005:**
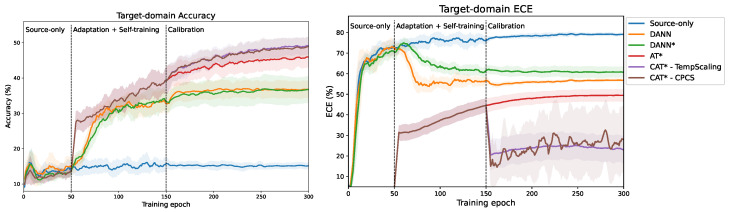
Evolution of target accuracy (**left**) and ECE (**right**) during training (here, on the 0→1 task in time domain). AT significantly improves over the DANN baseline. In addition, our proposed CAT effectively reduces calibration error, leading to an improved accuracy. Mean ± standard deviation over 5 runs with different random seeds. * = with MCC + SDAT.

**Figure 6 sensors-24-07539-f006:**
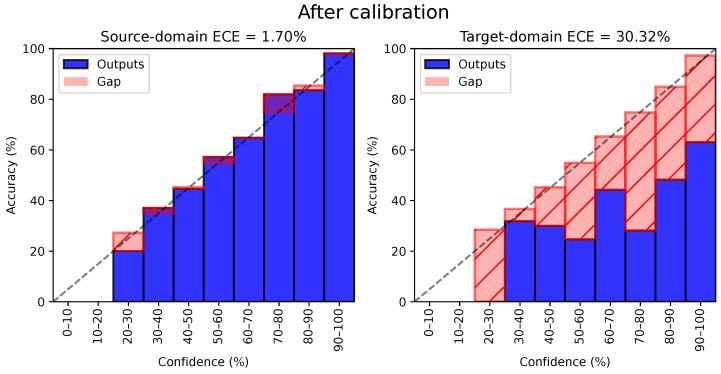
Example of reliability diagram after applying vector scaling calibration. Calibration improves in the source domain but not in the target domain. The dotted line represents ideal calibration.

**Figure 7 sensors-24-07539-f007:**
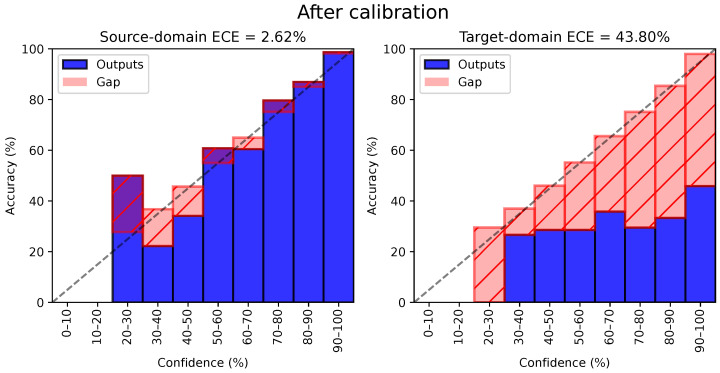
Example of reliability diagram after applying matrix scaling calibration. Calibration improves in the source domain but not in the target domain. The dotted line represents ideal calibration.

**Figure 8 sensors-24-07539-f008:**
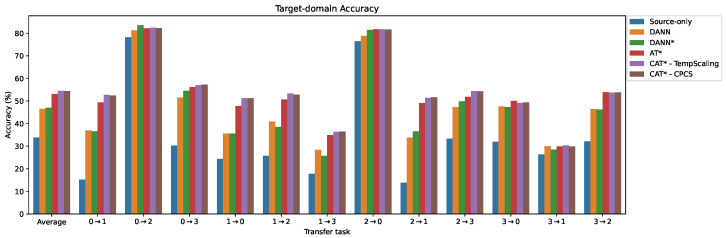
Comparison of target accuracy for different methods on the PU transfer tasks with time-domain input. * = with MCC + SDAT.

**Figure 9 sensors-24-07539-f009:**
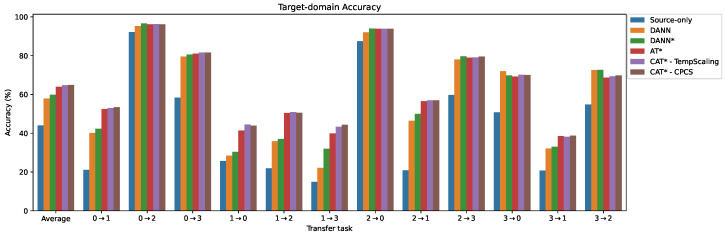
Comparison of target test accuracy for different methods on the PU transfer tasks with frequency-domain input. * = with MCC + SDAT.

**Figure 10 sensors-24-07539-f010:**
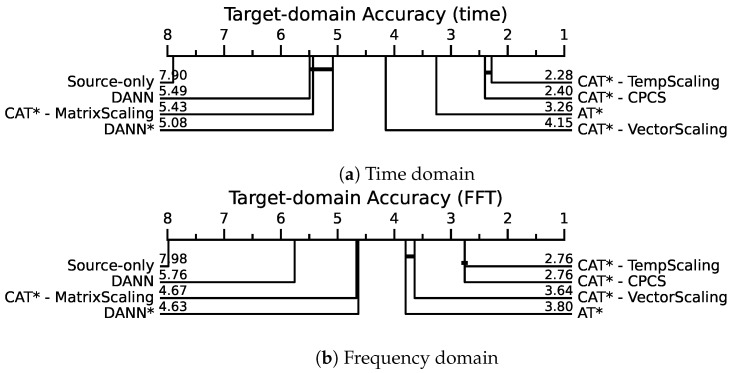
Critical difference diagrams of the performance of each method in terms of average rank of target accuracy on the PU transfer tasks. Methods connected by a horizontal bar are statistically equivalent (using Wilcoxon–Holm post hoc analysis at a 0.05 significance level). * = with MCC + SDAT.

**Figure 11 sensors-24-07539-f011:**
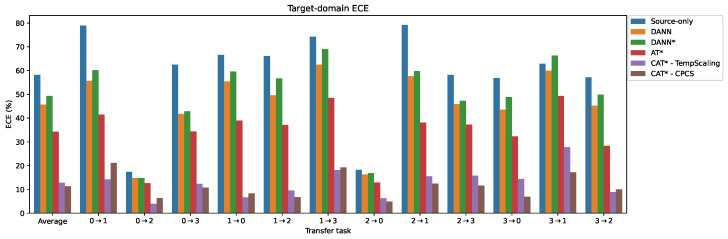
Comparison of target-domain expected calibration error (ECE) for different methods on the PU transfer tasks with time-domain input (lower is better). * = with MCC + SDAT.

**Table 1 sensors-24-07539-t001:** Comparison of self-training unsupervised domain adaptation methods applied to intelligent fault diagnosis.

Method	Backbone	Time	Frequency	Feature Alignment	Self-Training	Pseudo-Label Filtering	Auxiliary Loss
PCG-CNN [46]	1D-CNN	✓		-	MT with consistency loss	Fixed threshold	Class-balance loss
Wang et al. [49]	1D-CNN	✓		DAW	PL	Adaptive threshold	Triplet loss
DTL-IPLL [47]	1D-CNN	✓		MK-MMD	PL	Adaptive threshold + “make decision twice”	-
CAT (ours)	1D-CNN	✓	✓	DANN	MT with PL	Adaptive threshold + calibration	-

**Table 2 sensors-24-07539-t002:** Parameters of the CAT architecture used in this study.

Module	Layers	Parameters	Notation
1D-CNN backbone	Conv1D, BatchNorm, ReLU	in = 1, out = 16, kernel = 15	$W_{E}$
	Conv1D, BatchNorm, ReLU	in = 16, out = 32, kernel = 3	
	MaxPool1D	kernel = 2, stride = 2	
	Conv1D, BatchNorm, ReLU	in = 32, out = 64, kernel = 3	
	Conv1D, BatchNorm, ReLU	in = 64, out = 128, kernel = 3	
	AdaptiveMaxPool1D	out = 128 × 4	
	FC, ReLU	in = 128 × 4, out = 256	
	Dropout	*p* = 0.5	
Bottleneck	FC, ReLU	in = 256, out = 256	
	Dropout	*p* = 0.5	
Classification head	FC, Softmax	in = 256, out = 13	$W_{C}$
Domain classifier	FC, ReLU	in = 256, out = 1024	$W_{\mathrm{DC}}$
	Dropout	*p* = 0.5	
	FC, ReLU	in = 1024, out = 1024	
	Dropout	*p* = 0.5	
	FC, Sigmoid	in = 1024, out = 1	

**Table 3 sensors-24-07539-t003:** Domains for the Paderborn University (PU) dataset.

Domain	Rotational Speed	Load Torque	Radial Force
0	1500 rpm	0.7 Nm	1000 N
1	900 rpm	0.7 Nm	1000 N
2	1500 rpm	0.1 Nm	1000 N
3	1500 rpm	0.7 Nm	400 N

**Table 4 sensors-24-07539-t004:** Classification task for the Paderborn University (PU) dataset.

Class	Bearing Code	Damage	Element	Combination	Characteristic
0	KA04	Fatigue: pitting	OR	S	Single point
1	KA15	Plastic deform: indentations	OR	S	Single point
2	KA16	Fatigue: pitting	OR	R	Single point
3	KA22	Fatigue: pitting	OR	S	Single point
4	KA30	Plastic deform: indentations	OR	R	Distributed
5	KB23	Fatigue: pitting	IR (+OR)	M	Single point
6	KB24	Fatigue: pitting	IR (+OR)	M	Distributed
7	KB27	Plastic deform: indentations	OR+IR	M	Distributed
8	KI14	Fatigue: pitting	IR	M	Single point
9	KI16	Fatigue: pitting	IR	S	Single point
10	KI17	Fatigue: pitting	IR	R	Single point
11	KI18	Fatigue: pitting	IR	S	Single point
12	KI21	Fatigue: pitting	IR	S	Single point

**Table 5 sensors-24-07539-t005:** Target-domain test accuracy on the different PU transfer tasks with time-domain input. Average teacher accuracy over 5 runs (values in %).

Method	0→1	0→2	0→3	1→0	1→2	1→3	2→0	2→1	2→3	3→0	3→1	3→2	Average	Average Rank
Source-only ^†^	14.02	76.33	30.02	23.57	24.18	16.09	76.73	14.71	31.23	32.16	25.27	32.39	33.06	-
DANN ^†^	38.19	79.97	53.74	35.42	39.57	27.05	79.20	36.53	49.23	47.93	27.45	47.57	46.82	-
Source-only	15.15	78.23	30.29	24.33	25.71	17.73	76.41	13.87	33.34	31.92	26.32	32.12	33.78	7.90
DANN	36.96	81.22	51.56	35.55	40.89	28.35	78.77	33.83	47.29	47.56	29.94	46.41	46.53	5.49
DANN*	36.53	**83.54**	54.46	35.61	38.53	25.75	81.44	36.53	49.86	47.25	28.50	46.20	47.02	5.08
AT*	49.36	82.17	56.16	47.83	50.66	34.92	81.72	49.11	51.80	**50.08**	29.91	**53.86**	53.13	3.26
CAT*-TempScaling	**52.67**	82.44	57.00	**51.27**	**53.34**	36.34	81.78	51.44	**54.40**	49.22	30.37	53.74	**54.50**	**2.28**
CAT*-CPCS	52.42	82.29	**57.25**	51.24	52.76	**36.46**	81.69	**51.63**	54.25	49.40	29.88	53.77	54.42	**2.40**
CAT*-VectorScaling	34.51	82.08	50.95	43.66	45.53	34.10	**82.03**	40.18	48.35	47.99	**35.92**	46.75	49.34	4.15
CAT*-MatrixScaling	33.65	82.08	48.20	39.97	44.58	32.04	80.95	35.37	46.14	44.73	31.01	43.48	46.85	5.43

^†^ = results from [3] (Last-Mean). * = with MCC + SDAT.

**Table 6 sensors-24-07539-t006:** Target-domain test accuracy on the different PU transfer tasks with frequency-domain input. Average teacher accuracy over 5 runs (values in %).

Method	0→1	0→2	0→3	1→0	1→2	1→3	2→0	2→1	2→3	3→0	3→1	3→2	Average	Average Rank
Source-only ^†^	20.96	90.87	57.14	25.13	24.14	13.75	86.40	20.55	57.18	52.58	20.90	53.64	43.60	-
DANN ^†^	40.34	93.71	82.15	28.48	35.27	22.88	92.50	46.01	79.52	68.76	24.57	76.15	57.53	-
Source-only	21.13	92.12	58.34	25.68	21.92	14.92	87.47	20.83	59.76	50.81	20.71	54.75	44.04	7.98
DANN	40.09	95.27	79.58	28.45	35.88	22.15	92.07	46.44	78.03	**71.98**	32.09	72.61	57.89	5.76
DANN*	42.30	**96.58**	80.64	30.41	37.07	31.98	93.98	50.00	79.64	69.83	32.98	**72.73**	59.84	4.63
AT*	52.55	96.12	81.09	41.41	50.44	39.85	93.86	56.50	79.03	69.28	38.53	68.64	63.94	3.80
CAT*-TempScaling	52.98	96.27	81.57	**44.45**	**50.87**	43.36	93.89	**56.99**	79.06	70.11	38.22	69.34	64.76	**2.76**
CAT*-CPCS	**53.47**	96.18	**81.60**	43.96	50.60	**44.33**	93.89	56.93	**79.58**	70.05	**38.71**	69.77	**64.92**	**2.76**
CAT*-VectorScaling	50.92	96.40	81.24	43.53	48.40	41.97	**94.10**	55.18	78.61	69.25	37.55	69.50	63.89	3.64
CAT*-MatrixScaling	49.97	95.82	80.85	41.41	47.57	36.85	93.98	52.64	77.94	68.73	37.55	69.28	62.72	4.67

^†^ = results from [3] (Last-Mean). * = with MCC + SDAT.

**Table 7 sensors-24-07539-t007:** Target-domain expected calibration error (ECE) on the different PU transfer tasks with time-domain input. Average teacher ECE over 5 runs (values in %; lower is better).

Method	0→1	0→2	0→3	1→0	1→2	1→3	2→0	2→1	2→3	3→0	3→1	3→2	Average	Average Rank
Source-only	78.90	17.41	62.44	66.60	66.11	74.27	18.24	79.20	58.15	56.86	62.86	57.16	58.18	7.85
DANN	55.70	14.80	41.80	55.41	49.55	62.51	16.29	57.58	45.81	43.55	59.98	45.25	45.69	6.13
DANN*	60.17	14.81	42.89	59.57	56.73	68.97	16.86	59.73	47.24	48.83	66.33	49.83	49.33	6.88
AT*	41.51	12.60	34.33	38.96	37.04	48.46	12.90	38.07	37.27	32.34	49.30	28.29	34.26	4.15
CAT*-TempScaling	**14.16**	**3.95**	12.30	**6.64**	9.54	**18.10**	6.24	15.49	15.84	14.39	27.75	**8.87**	12.77	1.67
CAT*-CPCS	21.09	6.36	**10.76**	8.34	**6.77**	19.28	**4.89**	**12.44**	**11.63**	**6.94**	**17.19**	10.05	**11.31**	**1.48**
CAT*-VectorScaling	49.19	5.65	31.14	26.67	25.55	31.05	7.48	44.31	35.41	34.69	38.83	37.78	30.65	3.35
CAT*-MatrixScaling	54.38	8.03	36.62	27.25	25.97	32.55	11.29	48.81	41.30	39.78	47.16	42.60	34.64	4.48

* = with MCC + SDAT.

**Table 8 sensors-24-07539-t008:** Comparison between fixed and adaptive confidence thresholds in Adaptive Teacher (AT). The results on PU dataset with time-domain input. Average teacher accuracy on target test set over 5 runs (values in %).

Method	Threshold	0→1	0→2	0→3	1→0	1→2	1→3	2→0	2→1	2→3	3→0	3→1	3→2	Average
AT	Fixed	33.44	**80.95**	49.53	40.98	39.39	23.00	**79.75**	36.60	44.21	**50.23**	**29.20**	47.33	46.22
AT	Adaptive	**38.77**	80.89	**52.50**	**41.63**	**44.34**	**26.29**	78.86	**42.88**	**45.90**	49.03	28.99	**51.21**	**48.33**

**Table 9 sensors-24-07539-t009:** Ablation study on MCC and SDAT in Adaptive Teacher (AT) and our proposed Calibrated Adaptive Teacher (CAT) method. The results on PU dataset with time-domain input. Average teacher accuracy on target test set over 5 runs (values in %).

Method	MCC	SDAT	0→1	0→2	0→3	1→0	1→2	1→3	2→0	2→1	2→3	3→0	3→1	3→2	Average
AT			38.77	80.89	52.50	41.63	44.34	26.29	78.86	42.88	45.90	49.03	28.99	51.21	48.33
	✓		42.77	80.92	52.89	43.53	46.75	28.38	80.83	43.87	46.44	**51.34**	**32.73**	51.45	50.04
		✓	45.98	**82.32**	54.98	45.28	46.41	30.89	81.04	48.71	50.80	44.18	26.44	50.05	50.59
	✓	✓	**49.36**	82.17	**56.16**	**47.83**	**50.66**	**34.92**	**81.72**	**49.11**	**51.80**	50.08	29.91	**53.86**	**53.13**
CAT - TempScaling			44.58	81.40	53.95	47.04	46.99	27.87	78.92	44.82	48.14	49.59	28.99	52.18	50.28
	✓		46.56	81.34	55.16	48.91	50.14	32.44	81.17	46.38	48.35	**50.60**	**33.22**	53.07	52.28
		✓	49.14	82.41	56.58	47.50	49.92	32.47	81.01	50.00	52.34	44.70	23.19	50.14	51.62
	✓	✓	**52.67**	**82.44**	**57.00**	**51.27**	**53.34**	**36.34**	**81.78**	**51.44**	**54.40**	49.22	30.37	**53.74**	**54.50**
CAT - CPCS			43.90	81.53	53.86	46.18	46.44	27.35	79.17	44.29	48.53	49.62	30.15	52.40	50.28
	✓		46.87	81.34	54.89	48.82	50.38	32.95	81.17	46.78	48.59	**51.58**	**33.28**	52.89	52.46
		✓	49.82	**82.38**	57.19	47.37	49.65	30.98	81.01	51.23	52.68	45.19	23.40	49.71	51.72
	✓	✓	**52.42**	82.29	**57.25**	**51.24**	**52.76**	**36.46**	**81.69**	**51.63**	**54.25**	49.40	29.88	**53.77**	**54.42**

## Data Availability

Data and code are publicly available.

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
