# Peer review of "Calibrated Adaptive Teacher for Domain-Adaptive Intelligent Fault Diagnosis"

_sensors, 2024, doi:10.3390/s24237539_

Round 1
Reviewer 1 Report
Comments and Suggestions for Authors
In this paper, the Calibrated Adaptive Teacher (CAT) method is proposed to improve the quality of pseudo-labels and reduce error accumulation. the predictions of the teacher network on target samples throughout the self-training process is calibrated by leveraging well-known post-hoc calibration techniques. The paper demonstrates a reasonable approach and effective methods. Minor revisions are suggested for further clarification.
1. Please explain the reasons why each transfer task has different difficulty levels from the perspective of signal characteristics based on the selected experimental data. Suggest analyzing the impact of the characteristics of different transfer tasks on the effectiveness of the proposed method.
2. Why is CAT + post-hoc calibration (such as MatrixScaling) not as effective as DANN or AT in some transfer tasks? The positive effect of the post-hoc calibration has not been reflected. Could you provide more analysis of the reasons?
Author Response
Dear reviewer,
Thank you for your feedback. Please find our detailed response in the attached PDF document.
Sincerely

Reviewer 2 Report
Comments and Suggestions for Authors
Please refer to attached document for reviews

Author Response

(The authors gave the same response as above.)

Round 2
Reviewer 2 Report
Comments and Suggestions for Authors
Please find attached file of my reviews.

Author Response
Please find the response to the review for round 2 in the attached document.
Sincerely
